# Explanatory World Models via Look Ahead Attention for Credit Assignment

**Oriol Corcoll**[1]                    **Raul Vicente**[1]

[1]Institute of Computer Science, University of Tartu

## Abstract

Explanations are considered to be a byproduct of
our causal understanding of the world. If we would
know the actual causal relations, we could provide
adequate explanations. In contrast, this work places
explanations at the forefront of learning. We argue
that explanations provide a strong signal to learn
causal relations. To this end, we propose Explana-
tory World Models (EWM), a type of world model
where explanations drive learning. We provide an
implementation of EWM based on an attention
mechanism called look ahead attention, trained
in an unsupervised fashion. We showcase this ap-
proach in the credit assignment problem for re-
inforcement learning and show that explanations
provide a better solution to this problem than cur-
rent heuristics.

## 1 INTRODUCTION

> "Explanation is to cognition as orgasm is to
> reproduction"
>
> —Alison Gopnik [1998]

To the AI community, explanations may seem like a conve-
nient accessory lying on top of a magnificent and complex
cognitive system that communicates our inner thoughts and
representations to the outside world. [Gunning, 2017, Miller,
2018, Molnar, 2019, Lundberg et al., 2020]. In contrast, cog-
nitive and social scientists have considered explanations to
be central to human learning; they are not an accessory but
a major driver of learning. [Hempel and Oppenheim, 1948,
Gopnik, 1998, Keil, 2006, Lombrozo, 2006, Williams and
Lombrozo, 2013, Woodward and Ross, 2021]. Explanations
are regarded so crucial for learning that prediction and con-
trol are thought to depend on our capacity to explain events
and build relations between them [Lombrozo, 2011]. More-

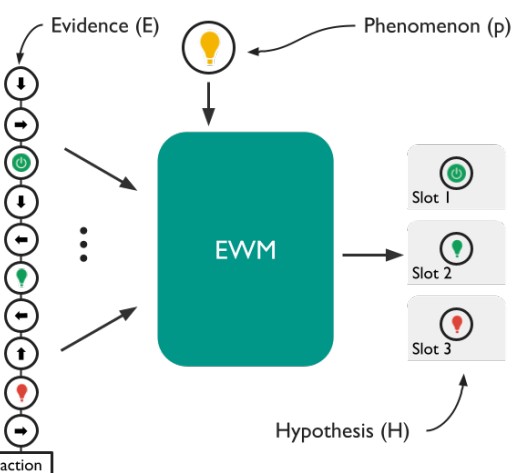

Figure 1: Explanatory World Models have two primary
sources of information: past events, or evidence ($E$), and an
event to be explained, the phenomenon ($p$). EWM explain a
given phenomenon by selecting $S$ past events as a hypothe-
sis ($H$) to why that phenomenon happened.

over, explanations are known to drive exploration in humans
[Legare, 2014], leading to the scientific method.

But, what is an explanation? Explanations can take many
forms and shapes, making them challenging to define [Lom-
brozo, 2006, Doshi-Velez and Kim, 2017]. For example,
Miller [2018] refers to explanations as an abductive process
that, given an event (phenomenon), can identify the causes
(explanans) that brought about the phenomenon. Similarly,
Elster [2007] characterizes explanations in the following
manner: "To explain a phenomenon (an explanandum) is
to cite an earlier phenomenon (the explanans) that caused
it". Accordingly, we propose Explanatory World Models
(EWM), Fig. 1, a type of world model that learns the en-
vironment's dynamics by explaining. In the context of re-
inforcement learning (RL), typical world models learn to
predict the future directly from a learned representation of
the past i.e. from causes to effects. Instead, our approach is

*Accepted for the Causal Representation Learning workshop at the 38th Conference on Uncertainty in Artificial Intelligence* (UAI CRL 2022).

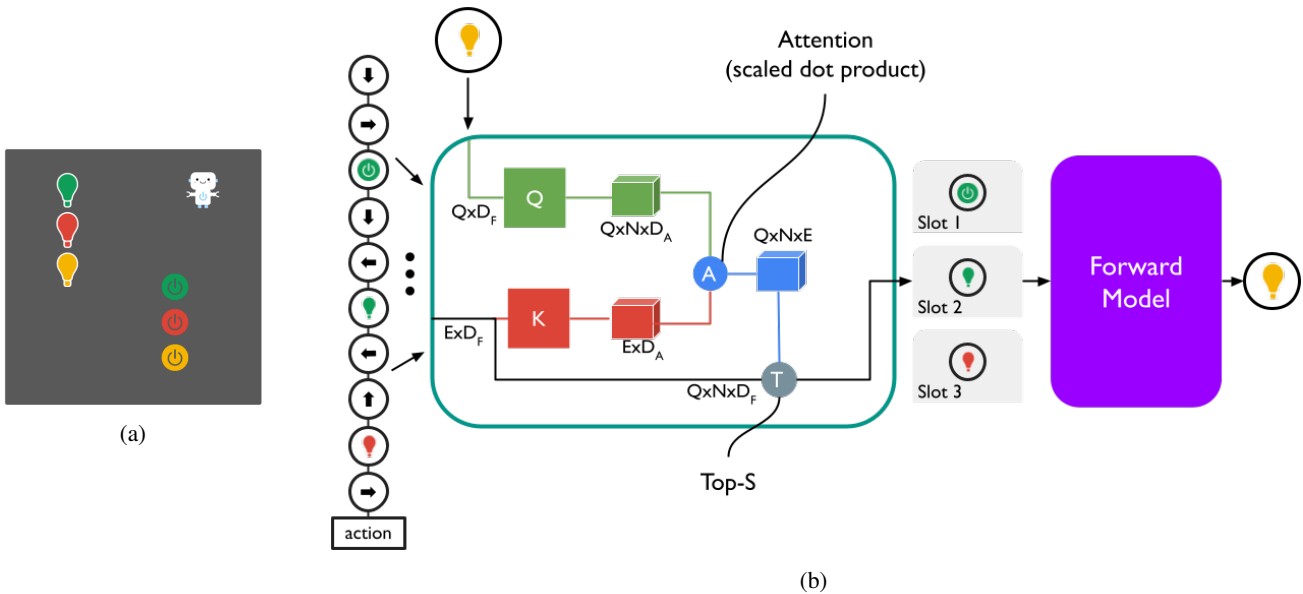

Figure 2: a) Lights environment where an agent has to turn on lights by pressing buttons. b) EWM based on Look Ahead Attention uses cross-attention between evidence and phenomenon to rank the evidence and selects the top-$S$ as hypothesis. The model is trained to predict the phenomenon only from the hypothesis.

to go on the opposite direction i.e. from effects to causes. In essence, EWMs relate a phenomenon to past events by creating a hypothesis as to what made the phenomenon happen.

In the following section, we characterize EWMs and propose an implementation based on attention [Vaswani et al., 2017]. An essential aspect of EWMs is their training; the specific definition of explanation will depend on the objective used. This work explores a predictive definition, where a satisfactory explanation is the one that allows predicting the phenomenon from the generated hypothesis. We showcase EWMs in the task of credit assignment, where a reward is redistributed to events controlled by the agent using the relations learned by the EWM.

## 2 EXPLANATORY WORLD MODELS

World models internalize the dynamics of the environment to enable planning, imagination or credit assignment. Typical approaches predict what will happen next from a learned representation of the state; in other words, these models associate past events with a distribution of possible future events. In contrast, if our goal is to learn the dynamics of the environment, would not be easier to go in the opposite direction and use the fact that an event happened to relate it to past events? Consider the environment in Fig. 2a where a light may turn on after 6, 7 or 8 time steps from pressing a button; knowing exactly when the light will turn on is an impossible task that forces the model to deal with this uncertainty. The real world has more complex relations that

make this task harder. Instead of dealing with a prediction problem, EWMs face a search problem which we argue is an easier task in complex environments.

We call this class of models, Explanatory World Models (EWM) since they are based on the abductive nature of explanations [Keil, 2006, Lombrozo, 2006, 2012, Miller, 2018]. Fig. 1 depicts the structure of EWMs. As with typical world models, EWMs rely on past events or evidence ($E$) but unlike them, EWMs must be provided with an event to be explained, the phenomenon ($p$). The phenomenon is used to select a subset of events as a hypothesis ($H$) for what made it happen.

Although this approach does not allow to plan or learn in an imagination space on its own, its use is twofold. First, by working backwards it enables to perform meaningful credit assignment. Most RL solutions to credit assignment rely on a temporal heuristic that gives more credit the closer an action is to a reward. This does not hold in complex environments. EWM use the learned relations to replace this heuristic, more on this in the following section. Second, although we do not explore this approach, planning and imagination could be enabled for EWM by predicting a high-level goal and generating hypothesis backwards, which we believe is an approach closer to how humans plan. Additionally, we do not conceive EWM as replacement to traditional world models; similarly to human common sense, EWM could provide constraints to these models.

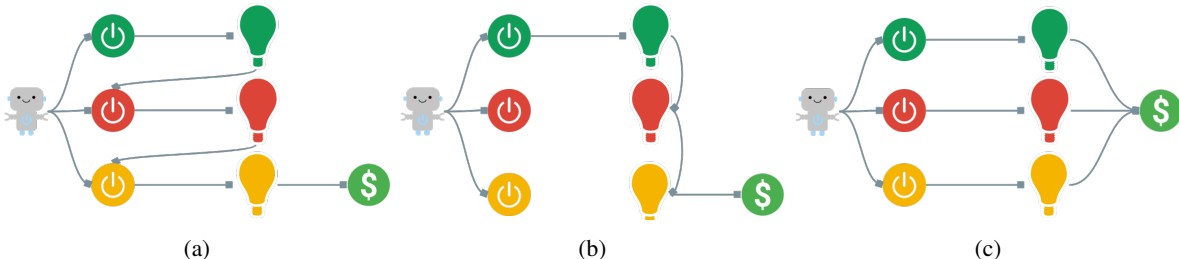

Figure 3: The agent needs to turn all the lights by pressing a series of buttons. The connection between buttons and lights is specified by a causal graph. a) **Chain:** each button activates a light but the red and yellow buttons are disabled until a light enables them. b) **One-turns-all:** the agent only needs to press the green button and wait until the lights turn on. c) **Independent:** each light is turned on independently.

## 2.1 ATTENTIONAL EWM

In the following, we introduce an implementation of EWM based on attention [Vaswani et al., 2017], depicted in Fig. 2b. The goal of an EWM is to create a hypothesis to why a phenomenon happened. Similarly to the approach taken by Tang et al. [2020], we propose to rank the evidence based on how well it explains a phenomenon using the scale dot product attention. Then it selects the $S$ most relevant events as a hypothesis, similarly to recent Mixture-of-Experts models [Fedus et al., 2021, Riquelme et al., 2021] or the RIMs framework [Goyal et al., 2020]. The ranking of evidence is done using cross-attention between phenomenon and evidence, where the phenomenon is used to compute queries and evidence is used to compute the keys. Note that the value function is set to the identity. We use the differentiable top-$S$ function to select the most informative events as hypothesis.

It is crucial to notice that we cannot directly optimize this model; left aside that explanations are hard to define, there may be many valid hypotheses, and ground truth for these may be hard to collect. Instead, we propose to use a predictive view of explanations that allows us to build an objective that can be optimized for. We say that a hypothesis explaining a phenomenon is satisfactory if it allows predicting the phenomenon only using the hypothesis. Therefore, we introduce a forward model to predict the phenomenon solely from the hypothesis.

Note that since we use the phenomenon in both, the objective and the input, the model could find trivial solutions. In other words, the model could encode the phenomenon in the hypothesis directly, without using the semantic content of the evidence. Allowing only $S$ number of events in the hypothesis creates a second bottleneck that makes easier to encode the right semantic information than just "leaking" the phenomenon. Since we allow the model to use the future, i.e. the phenomenon, for ranking but not for prediction, we call this approach look ahead attention.

**Events:** In this work we define an event as the delta between observations $e = o' - o$. Ideally, we would want events to be as independent and disentangled as possible i.e., if a piece of evidence contains multiple objects, the model will assign all these objects as hypothesis. Learning disentangled representations is an important line of research that we hope to see evolve in the near future but that we circumvent by having an environment where only one event at every time step can happen. As in Ramesh et al. [2021], it would be prohibitive for the EWM to work directly with pixels, thus we first use a Discrete VAE to create a low-dimensional representation of each event before passing it to the EWM. The discrete codes are then transformed into a continuous representation using linear embeddings. Additionally, we add relative positional encodings [Dai et al., 2019] to events.

**Controlled events:** An important factor when planning, imagining or doing credit assignment; is the ability to know what changes in the environment were caused by the agent. This is particularly important in the task of credit assignment where reward must be given only to actions that made the reward happen. To achieve this, we include the action performed by the agent as an additional piece of evidence. As done in Corcoll and Vicente [2020, 2021], if the model needs to use the action to predict the phenomenon, we say the phenomenon was controlled by the agent. Actions are transformed using a linear embedding layer to match the size of the rest of events. Note that special care needs to be taken when computing the temporal mask since we want to only reveal the last action taken. Similarly, the positional encoding for each action must be the same, regardless of where in the sequence it is.

**Training and setup:** We implement the Discrete VAE using convolutional layers. The query and key transformations of the look ahead attention module are two-layer MLPs with ReLU activations. Similarly, the forward model is a three-layer feed forward network with ReLU activations. We first train the Discrete VAE in isolation with samples collected by a random policy. Then, the attentional EWM and forward model are trained end-to-end to predict the discrete codes of the phenomenon produced by the frozen Discrete VAE.

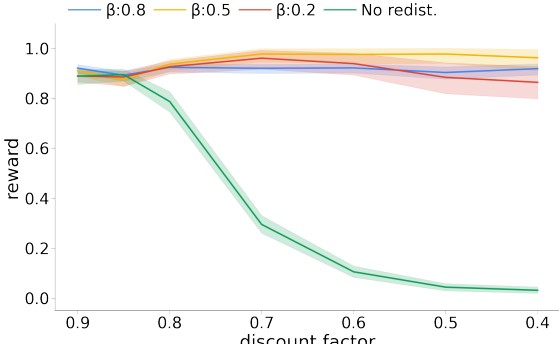

Figure 4: Vanilla PPO (green) stops solving the task when the discount factor is lower than $0.8$. Using EWM to redistribute reward allows the agent to learn even with low discount factors and low redistribution factor ($\beta$).

## 3 REWARD REDISTRIBUTION

An arguably underexplored area of RL is credit assignment. The predominant approach to this problem is to propagate reward based on a simple heuristic: the closer the action is to the reward, the more important it is for that reward to happen. Unfortunately, this approach only works in environments with dense rewards but fails in sparse reward environments where temporal correlations between events do not work, making the learning highly inefficient.

Instead, explanations create a relational directed graph between events where the immediate neighbors of a phenomenon are events in the hypothesis. We can exploit this graph and redistribute reward to events that made the phenomenon happen. Since our implementation takes the top-S events, we remove neighbors with attention weights lower than a set threshold. In RL we typically reward actions, not events; thus, reward is redistributed only to events controlled by the agent (i.e., events where the action is part of the hypothesis). Moreover, we can iteratively explain events in the hypothesis and redistribute reward accordingly. Note that this approach allows us to redistribute reward over arbitrarily long horizons, the only requirement is that relations between two events fall in the same attention window. In practice, we limit how deep we go into the graph by reducing the reward at each level and stop the redistribution if the reward is below $0.1$. Finally, the reward to the action of each controlled event is given by

$$r_i = \beta^i r \qquad (1)$$

where $\beta$ is the redistribution factor between $[0, 1]$ and $i$ indicates the level in the graph. The higher the redistribution factor the deeper in the graph the reward will travel.

## 4 EXPERIMENTS

Our experiments explore the following questions. 1) can these relations be used to redistribute credit so as to achieve more efficient learning? and, 2) can the Attentional EWM learn the right relations between events under different underlying dynamics?

For all our experiments we use the Lights environment Fig. 2a where an agent needs to turn on the lights by pressing a set of buttons. To evaluate how well EWM deal with delays between events, a light will turn on 7 steps after pressing the corresponding button or the previous light turns on. The connectivity between lights and buttons is determined by a causal graph. In some cases, buttons may be disabled until a light is turned on. The three different causal graphs used in this work are shown in Fig. 3. In these experiments we train a EWM for each variant with samples collected using a random policy.

### 4.1 CAN EWM PROVIDE A MORE EFFICIENT LEARNING?

Here, we want to analyze if redistributing credit using the EWM provides a more efficient learning than traditional credit assignment. In this experiment, we train a PPO agent with and without redistribution using different discount factors and redistribution factors on the Chain variant Fig. 3a). Fig. 4 shows how different discount and redistribution factors affect the learning. A vanilla PPO agent has a hard time solving the Chain variant when the discount factor $\gamma$ reaches $0.8$. On the other hand, our redistribution method allows the agent to learn even when both, discount and redistribution factors are low.

### 4.2 DOES EWM REDISTRIBUTE REWARD CORRECTLY?

Here we analyze the relations learned by the EWM and how the reward is propagated in a single episode after training the agent. The following results use a PPO agent with discount factor $0.8$ and (if applicable) redistribution factor of $0.8$.

#### 4.2.1 Chain variant (Fig. 3a):

Fig 5 presents the results on the chain variant, blue line denotes the learned value function at the end of training and green line is the reward assigned to each time step.

On the left, a PPO agent trained without any reward redistribution and discount factor of $0.8$. Vanilla PPO fails to propagate reward to far enough events. Relevant events happening far from the reward get little attention and irrelevant events close to the reward too much. Traditional methods have a hard time propagating reward to the right events.

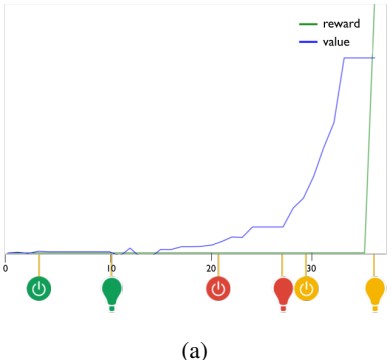

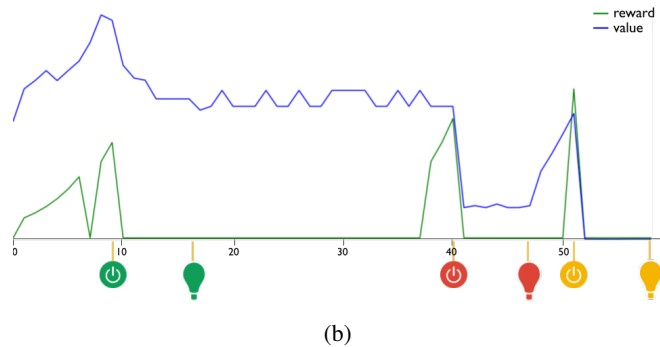

|     |     |
|:---:|:---:|
| (a) | (b) |

Figure 5: PPO without (a) and with (b) reward redistribution after training on the chain variant. The EWM model can identify the right relations between events and redistribute the reward to key actions.

In contrast, the plot on the right shows a PPO agent trained with the reward redistribution method proposed in section 3 and discount factor of $0.8$. Here the right events (pressing buttons and moving to those buttons) are rewarded.

### 4.2.2 One-turns-all variant (Fig. 3b):

This variant has long delays between action and reward. Here, the agent only needs to press the green button and wait until every light turns on. Pressing other buttons does not have an effect on the lights nor reward.

As shown in Fig. 6a, the model learns the right relations and reward is not given to pressing other buttons nor to the lights turning on since these are not directly controlled by the agent. The model learns to reward the button that causes the lights to turn and the movement of the agent necessary to get to the button.

### 4.2.3 Independent variant (Fig. 3c):

Here each light is turned on independently and only once all lights are on the reward is given. The complexity of this environment is to relate multiple events to the reward. In previous variants, the reward was always given when the yellow light turned on. Note that here the event of getting a reward can happen when any of the lights turns on but the reward is caused by all the lights turning on.

The results in Fig. 6b show that the EWM can correctly identify that the yellow, red and green buttons caused the reward to happen. Unfortunately, the red and yellow buttons are not in the direct neighborhood of the reward since the model needs to go deeper in the graph (note that the height of the reward indicates the depth in the graph).

## 5 RELATED WORK

**Credit assignment:** multiple methods have been devised to improve the credit assignment problem: RUDDER[Arjona-Medina et al., 2019], TVT [Hung et al., 2019], SECRET [Ferret et al., 2020] and Synthetic Returns[Raposo et al., 2021]. All these methods learn relations between events and reward. A drawback of relying on reward to build a model of the world is that there is no learning until a reward is discovered. This is counterproductive since the dynamics of the world are typically stable and can be learned even without seeing any reward. EWM learns how events relate to each other and uses the learned relations to propagate reward to the causing actions.

**Slots:** RIMs framework [Goyal et al., 2020] and Slot-Attention[Locatello et al., 2020] learn to route information to slots. By limiting the number of evidence the forward model can use, EWM creates an additional bottleneck.

## 6 CONCLUSIONS

In this work, we have argued that explanations are not an accessory to learning but a central component. To this purpose, we have introduced a class of models called Explanatory world Models (EWM) and proposed an implementation based on attention. We showcased EWM in the task of credit assignment where our experiments demonstrated that this approach can greatly reduce the sample inefficiency of traditional approaches.

In future work, we want to expand EWM to environments with more complex dynamics. Another open question is how can events be learned in these complex environments? Additionally, explanations are know to drive our curiosity, would agents rewarded by finding phenomenons hard to explain explore better? Finally, can world models for planning or learning in imagination benefit from obeying constraints imposed by a EWM?

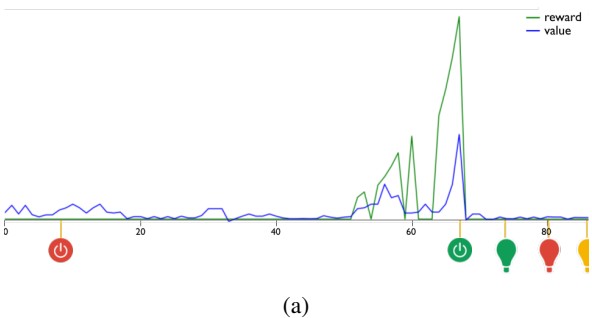 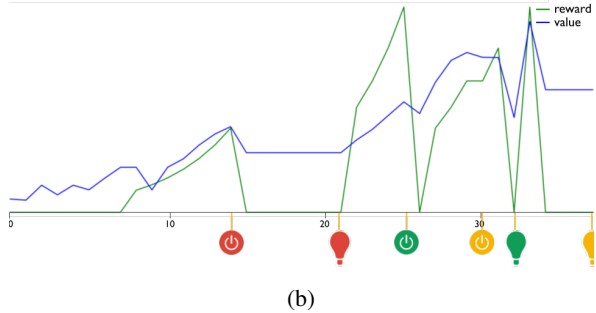

(a)  (b)

Figure 6: a) Reward is not redistributed to non-causal events (red button). Moreover, reward is also not given to uncontrolled events (lights). b) The EWM is able to identify that pressing all the buttons was what caused for receiving a reward.

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
