# OpenReview forum: "Explanatory World Models via Look Ahead Attention for Credit Assignment"
_auai.org/UAI/2022/Workshop/CRL — CRL@UAI 2022 Poster_

### Official Review · Reviewer_zuLU · 2022-06-30
**Interesting idea but lack of convincing experiments**

**Rating:** 6
**Confidence:** 3

**Review:**

The main tenet of the authors work is that explanations are not just a post-hoc tool for building trust but a fundamental component of learning. The authors propose explanation based world models which they call explanatory world models (EWMs). In EWMs, the event or phenomenon that occurs is given as input. There is a candidate set of evidence that the phenomenon is compared against to arrive at a hypothesis explaining what caused the event.
I think the work is interesting and has reasonable degree of originality. However, the work does not do a good job of convincing the reader that explanations are fundamental to learning. I hope in a longer version of the work the authors can explore the task of credit assignment extensively and provide a more convincing case that explanations lead to better credit assignment, which further improves downstream performance. I found the writing of the work clear enough. Overall, I think the work should explore their claims more thoroughly in a longer version.

---

### Meta-Review · Program_Chairs · 2022-07-06

**Recommendation:** Accept (Poster)
**Confidence:** 3

**Metareview:**

The work was deemed interesting and original. The reviewer raised some concerns which the authors are encouraged to elaborate upon for a future version, and thoroughly address in a longer version of the paper.

---

### Decision · Program_Chairs · 2022-07-06

Accept (Poster)